# Differences by School Location in Summer and School Monthly Weight Change: Findings from a Nationally Representative Sample

**DOI:** 10.3390/ijerph182111610

**Published:** 2021-11-04

**Authors:** Ethan T. Hunt, Bridget Armstrong, Brie M. Turner-McGrievy, Michael W. Beets, Robert G. Weaver

**Affiliations:** 1Michael and Susan Dell Center for Healthy Living, University of Texas Health Science Center at Houston, Austin Campus, Austin, TX 78701, USA; 2Department of Exercise Science, University of South Carolina, Columbia, SC 29208, USA; ba12@mailbox.sc.edu (B.A.); beets@mailbox.sc.edu (M.W.B.); weaverrg@email.sc.edu (R.G.W.); 3Department of Health Education & Behavior, University of South Carolina, Columbia, SC 29208, USA; brie@sc.edu

**Keywords:** obesity, geography, children, vacation

## Abstract

Objectives: To examine changes in accelerations of Body Mass Index (BMI), age-and-sex specific body mass index (zBMI), and 95th percentile of BMI (%BMIp95) during the summer months and school year by school location designation (i.e., urban, suburban, exurban). This study utilized the Early Childhood Longitudinal Study Kindergarten Class of 2010–2011. Methods: Of the 18,174 children in the ECLS-K:2011 dataset, I restricted participants to those with at least two consecutive measures that occurred August/September or April/May. Mixed-effect regression analyses estimated differences in monthly change in BMI, zBMI, and %BMIp95 between the summer and school year while accounting for the ECLS-K complex sampling design. Models also examined differences in the magnitude of BMI, zBMI, and %BMIp95 change between the summer and school year by school location. Post-hoc Benjamini–Hochberg (BH) procedure set at 10% false discovery was incorporated to account for multiple comparisons. Results: A total of 1549 children (48% female, 42% White) had at least two consecutive measures that occurred in August/September or April/May. Among all locale classifications (i.e., urban, suburban, and exurban), children from high-income households comprised the largest proportions for each group (31%, 39%, and 37%), respectively. Among urban and suburban locations, Hispanic children comprised the largest proportions for both groups (43% and 44%), respectively. Among exurban locale classifications, White children comprised the largest proportion of children (60%). Children from suburban and exurban schools experienced significantly less accelerations in monthly zBMI gain when compared to their urban counterparts −0.038 (95CI = −0.071, −0.004) and −0.045 (95CI = −0.083, −0.007), respectively. Children from exurban schools experienced significantly less acceleration in monthly %BMIp95 during the summer months when compared to the school year −0.004 (95CI = −0.007, 0.000). Conclusions: This is one of the first studies to examine summer weight gain by school location. Summer appears to impact children more negatively from urban schools when compared to their suburban and exurban counterparts.

## 1. Introduction

Childhood obesity is a significant public health concern [1]. Obesity is linked to various chronic diseases and disorders such as heart disease, cancer, insulin resistance, and cardiovascular disease [2,3]. Where children live and attend school can be significant contributors to obesity [4]. In recent years, there has been a substantial body of research examining the relationship between school location (i.e., urban, suburban, exurban) and risk for childhood obesity [5]. 

While there are many ways to define locality, the United States (U.S.) Census defines urbanized areas as areas with a population of 50,000 or more, either adjacent to densely populated areas or other areas close in proximity with densely populated areas [6]. Areas directly outside urban areas are defined as suburbs. Suburbs include incorporated areas or census designations with at least 2500 inhabitants, usually outside of urban areas. Exurban (i.e., town and rural locations) areas encompass localities outside of both urban and suburban areas [7]. Children who live in exurban settings are significantly more likely to have overweight/obesity when compared to their urban counterparts [4,8]. One recent meta-analysis concluded that children (2–19 years) in exurban settings have more than 25% greater odds of obesity compared to urban children [5]. 

There is a limited amount of research that explicitly examines childhood obesity prevalence in suburban settings. The limited research indicates that both females and males [9] in suburban areas have lower rates of obesity compared to their same-sex peers in urban or exurban settings. The underlying mechanisms driving these differences are likely multifaceted. Counterintuitively, it appears that children in rural settings engage in more physical activity (PA) than their urban counterparts, despite having higher rates of obesity [10,11,12]. In addition, differences in access to community resources may play a role. Specifically, more deprived exurban areas have less access to community resources such as healthcare and recreational facilities, supermarkets, and other essential services compared to their urban and suburban counterparts [13,14,15]. 

Summer vacation from school may exacerbate disparities in access to resources for children in exurban areas compared to urban and suburban children. For instance, food insecurity is a risk factor for obesity in children [16]. During summer vacation, food security decreases in states providing small numbers of Summer Food Service Program meals and summertime school lunches than in other states [17]. Although the United States Department of Agriculture’s Summer Food Service Program provides free meals for at-risk children during the summer [18], not all children can access this service. Lack of transportation and long distances to program sites are barriers to children in exurban areas accessing these services [19]. Still, they may not be a barrier for children in more densely populated urban and suburban areas. These findings suggest that the removal of school during the summer may partially explain disparities in childhood obesity that exist by geographic location.

A large body of evidence indicates children’s body mass index (BMI) accelerates, and the prevalence of children with obesity increases during the months of summer [20,21,22,23,24,25]. Previous studies have shown that Black children and children from low-income households experience greater summer BMI gain than their White and high-income counterparts [22,26]. These disparate rates of accelerated summer BMI gain may also contribute to the disproportionately high rates of overweight and obesity observed among Black and low-income children [27,28]. Disparate rates of overweight and obesity by locality may be similarly explained by disproportionate rates of accelerated summer BMI gain. However, no studies to date have examined rates of children’s accelerated summer BMI gain by locality. Thus, the current study aimed to investigate accelerations in body composition during the summer months by school locality (i.e., urban, suburban, exurban). 

## 2. Materials and Methods

This project utilized publicly available data from the Early Childhood Longitudinal Study, Kindergarten Class of 2010–11 (ECLS-K:2011) [29]. ECLS-K:2011 is a complex multistage probability sample from the population of U.S. children who were enrolled to attend kindergarten in the fall of 2010. ECLS-K:2011 is designed to provide data on children’s early school experiences beginning in kindergarten and following through fifth grade. Sampling for participants followed a multistage design, which took place in three primary steps. The United States was divided into Primary Sampling Units (PSUs), and 90 of these PSUs were sampled in the first step. Second, public and private schools were sampled within each of the 90 PSUs. Third, children enrolled in kindergarten programs in those schools were sampled [29]. Each primary sampling unit was a large county or group of adjacent and demographically similar small counties [29]. The ECLS-K:2011 includes both public and private schools. Asian/Pacific Islander children were over-sampled to assure that the sample included enough students of this race/ethnicity to make accurate estimates for these participants as a group [29]. Sampling weights were used to adjust for differential probabilities of selection at each sampling strata and adjust for the effect of nonresponse on parameter estimates [30]. The complete dataset consists of 18,174 children in 970 schools and 90 primary sampling units. 

ECLS-K:2011 collected fall measurements from August to December and spring measurements from March to June [22]. A measurement occasion (i.e., month height/weight was collected) variable was included in the ECLS-K:2011 dataset. Because we aimed to estimate monthly school and summer changes in body composition, we restricted our analytical sample to children measured in August/September and April/May. This approach allowed us to examine differences in body composition change during school (more-structured) and summer (less-structured) without employing analytical methods that would predict all children’s height/weights at one specific time-period. After restricting the dataset to the defined months, we had an analytical sample of 1549 children. Figure 1 shows the flow of exclusions from the complete sample to the analytic sample.

### 2.1. Measures

#### 2.1.1. Body Composition

On each measurement occasion, height and weight were measured twice, using a Shorr board stadiometer and a Secca Bella 840 flat electronic scale [29]. Children were asked to remove shoes, hats, and jackets [29]. BMI was calculated by first taking the average of the measures, then using the standard formula BMI = kg/m^2^. zBMI score was then transformed into age-and-sex specific zBMI score [31]. In addition, we calculated percentage of the 95th percentile of BMI (%BMIp95). %BMIp95 is the ratio of individual BMI relative to the sex- and age-specific Center for Disease Control and Prevention 95th percentile BMI multiplied by 100. Thus, a %BMIp95 > 100% indicates that an individual child is obese [32]. %BMIp95 is included as an additional measure as it may be more appropriate than zBMI change for tracking change in age and sex specific BMI over time, especially for those children with extreme zBMI values and does not differ solely due to differences in sex and age as zBMI may [33]. 

#### 2.1.2. Location 

To measure school locality, ECLSK:2011 categories correspond to the 2006 system National Center for Education Statistics (NCES) location variables constructed in combination with the United States Census Bureau classification system [6,34]. Geocoding technology and the Office of Management and Budget defined metro areas that rely less on population size and county boundaries than the proximity of an address to an urbanized area [35]. NCES school location designations and definitions are as follows: city, suburb, town, and rural. Cities consist of areas inside an urbanized area and inside a principal city (i.e., the core city in a metropolitan area) with a population of 250,000 or more. Suburbs consist of areas outside a principal city and inside an urbanized area with a population of 250,000 or more. Towns consist of areas inside an urban cluster less than or equal to 10 miles from an urbanized area. Rural areas consist of census-defined rural territory that is less than or equal to 5 miles from an urbanized area and rural territory that is less than or equal to 2.5 miles from an urban cluster. For this study, we chose to categorize school locations as urban, suburban, and exurban, in that schools in “towns”, and “rural settings” are defined as exurban [7].

### 2.2. Covariates

#### 2.2.1. Race/Ethnicity

Race and ethnicity of children were measured via parent proxy report. Consistent with previous work using ECLSK:2011 datasets, [36,37] we designated children into five racial/ethnic categories (i.e., non-Hispanic White, Black, Hispanic, Asian, and “Other”). Children whose parent/guardians designated them as non-Hispanic White were categorized as White. Children whose parent/guardians defined them as non-Hispanic Black were classified as Black. Children whose parent/guardians designated them as Hispanic were categorized as Hispanic. Children whose parent/guardians designated them as Asian or Pacific Islander were categorized as Asian. All other children whose parents identified them as any other racial/ethnic category other than non-Hispanic White were classified as Other. 

#### 2.2.2. Income-to-Poverty Ratio

Parent/guardians reported their total household income for the previous year to the nearest thousand during the parent interview portion of data collection [29]. The income-to-poverty ratio was calculated by dividing the total reported household income by the Department of Health and Human Services’ poverty threshold [38], where lower scores represent greater unmet need. For example, a poverty-to-income ratio (PIR) of 0.5 indicates that a household is earning an equivalent of 50% of the income as the federally established poverty threshold. In contrast, an income-to-poverty ratio of 1.5 indicates that a household is earning 150% of the federally designated poverty threshold. For this study, each participant was classified as above 200% PIR and below 200% PIR based upon their household income-to-poverty ratio. Classifications aligned categories defined by the ECLS-K:2010–11 database [36] and were created as follows. If a child’s household received an income-to-poverty ratio of 0.00–1.99, the child was classified as below 200% PIR. Children living in households with an income-to-poverty ratio of >2.00 were classified as above 200% PIR.

#### 2.2.3. Parental Employment Status and Education

Parent/guardians reported their employment status for the previous week during the parent interview portion of data collection [29]. Employment status was specific to “job for pay” in that parents were asked a dichotomous (yes or no) during the past week, “did you work at a job for pay?” Similarly, parent/guardians reported their highest education received. Education was categorized as the following: high school, but no diploma, high school diploma, vocational/technical program after high school, some college but no degree, associate degree, bachelor’s degree, professional degree after bachelor’s degree, or “other”.

### 2.3. Analysis

All statistical analyses were performed using Stata software version 16 (StataCorp, College Station, TX, USA). After restricting the sample to children with measurements in August/September for fall and April/May for spring, descriptive statistics were calculated for the full sample of participants and the restricted analytical dataset. 

Multiple imputation was performed to estimate missing values to account for the potential bias introduced by large amounts of missing data in analyses of complete-case data [39]. Specifically, missing covariate values of poverty status and parental education were imputed using the chained equations methodology and generated ten imputed data sets [40]. Poverty status (via poverty threshold) and parental education were chosen to examine because descriptively, it was shown to have high proportions of missing within these covariate variables (>25%). To do this, a stepwise approach occurred as follows. First, because of the large dataset, missing cases of covariate data were identified by performing a missing value analysis pattern test by employing Little’s missing cases at random (MCAR) test [41,42]. Using Stata, the mcartest command implements the chi-square test of MCAR, which tests whether significant differences exist between the means of different missing-value patterns. The test statistic takes a form similar to the likelihood-ratio statistic for multivariate normal data and is asymptomatically chi-square distributed under the null hypothesis that there are no differences between the means of different missing-value patterns. Rejection of the null provides sufficient evidence to indicate the data are MCAR [42]. The relationship between all covariates in the analytical models was explored via a chi-square test. These tests indicate that other covariates included in analyses (i.e., school designated location and race/ethnicity) were related to poverty status and parental education. Relationships of covariate variables are as follows: (location vs. parental education, *p* < 0.00), (sex vs. parental education, *p* = 0.20), race/ethnicity vs. parental education, *p* < 0.00), (location vs. poverty status, *p* = 0.02), (sex vs. poverty status, *p* = 0.18), and (race/ethnicity vs. poverty status, *p* < 0.00).

After examining descriptive data and missingness utilizing the Little MCAR and chi-square test, it was determined that parental education and poverty status were not MCAR. Thus multiple imputation of said missingness would be conducted prior to final analyses. This was done by incorporating multiple imputation while accounting for the complex survey design using study weights. The included weighting variable accounted for nonresponse at the school, child, and parent level. The ECLSK analytical handbook recommends incorporating a weighting variable that includes the most cases to produce estimates that represent the cohort of children in kindergarten in 2010–2011.

For this reason, the weighting variables W2SCH0 and W7CF7P_7 were included. W2SCH0 accounts for nonresponse at the school level. W2SCH0 encompasses the child base weight adjusted for nonresponse associated with child assessment/child questionnaire data from all seven rounds from kindergarten through third grade and parent data from all seven rounds from kindergarten through third grade. Multiple imputation in Stata involves three steps. First, Stata is informed that multiple imputation will be conducted by setting the dataset. Next, it is indicated which variables with missing observations will be imputed by registering them. To reproduce these results, Stata recommends setting a seed with a random number; otherwise, Stata will draw different samples every time it runs the imputation procedure, and replication will not be possible. Finally, a specified number of times the missing values should be replaced is conducted, producing ten datasets while accounting for weighting variables using weights that include adjustments for nonresponse. All of the datasets were combined into one single multiple-imputation dataset. Finally, the imputed datasets are now available to estimate various models, including regression models. To run a regression model using this imputed dataset, stata instructs to add the following rule (mi estimate, dots) before the command. When initiating an analysis, Stata now produces an output for the pooled dataset with ten imputations. Following imputation, separate multilevel mixed-effects linear regressions nested at the child level estimated differences in monthly BMI, zBMI, and %BMIp95 change. Monthly change scores for the school year BMI, zBMI, and %BMIp95 were created by subtracting a child’s fall measure from the spring and dividing by the number of months between each child’s fall and spring measure. The same process for the summer by subtracting the spring measure from the subsequent fall measure and dividing by the number of months between each child’s spring and subsequent fall measure. Monthly change was selected as the outcome measure to account for the difference in time period for a school year (i.e., 9-months) and summer (i.e., 3-months). In addition to the dependent variable (monthly BMI, zBMI change, or %BMIp95), separate models were constructed to examine the impact of monthly change in body composition by our independent variables: race/ethnicity, poverty status, and location. Each separate model then included post-estimation interactions that allowed us to examine within-group summer vs. school change and between-group difference in summer change using the lincom post-estimation procedure for multilevel and longitudinal modeling in Stata [43]. All models included sex, age, parent’s employment status, parent’s education, race, and poverty-to-income ratio as covariates. Finally, to correctly estimate variance, taking into account the clustered, multistage sampling design and the use of differential sampling rates to oversample targeted subpopulations, standard errors were estimated using the Taylor series method and weighting variables were included [44]. The same sampling weights used in the multiple imputation were included in all models [29]. Finally, a post-hoc Benjamini–Hochberg (BH) procedure set at 10% false discovery was incorporated to account for multiple comparisons [45].

## 3. Results 

Demographics and sample sizes are presented in Table 1. A total of 1532 children (52% female, 42% White) were included in the final analytic sample. Raw monthly changes in BMI, zBMI, and %BMIp95 during the school year and summer months are presented in Table 2. Model-based estimates examining within and the between-group difference in change by rurality can also be found in Table 2. Model-based within-group differences, examining differences between monthly changes during the school year and summer months by BMI, zBMI, and %BMIp95, increased by 0.095 (95CI = 0.074, 0.117), 0.016 (95CI = −0.004, 0.027), and 0.001 (95CI = 0.000, 0.002) more during the summer compared to the school year, respectively. For children from urban schools, school change in raw BMI, zBMI, and %BMIp95 was 0.028 (SD = 0.193), 0.002 (SD = 0.074), and −0.001 (SD = 0.010), respectively. For children in urban schools, summer change in raw BMI, zBMI, and %BMIp95 was 0.071 (SD = 0.230), 0.007 (SD = 0.126), and 0.000 (SD = 0.012), respectively.

For children from urban schools, model based within group differences during the summer reached statistical significance by BMI, and %BMIp95 at 0.083 (95CI = 0.049, 0.117), and 0.004 (95CI = 0.000, 0.006), respectively. For children from suburban schools, raw school change in BMI, zBMI, and %BMIp95 was 0.027 (SD = 0.158), 0.006 (SD = 0.076), and −0.001 (SD = 0.008), respectively. For children from suburban schools, summer change in raw BMI, zBMI, and %BMIp95 was 0.088 (SD = 0.202), 0.012 (SD = 0.098), and 0.001 (SD = 0.010), respectively. For children from suburban schools, within group differences in accelerations during the summer were statistically significant for BMI during the summer months 0.047 (95CI = 0.004, 0.090), but not zBMI, and %BMIp95, 0.008 (95CI = -0.013, 0.029), and 0.002 (95CI = −0.000, 0.004), respectively. School location did not predict significant between group differences in BMI change during the summer, indicating that children’s summer accelerations in BMI were not significantly different by school location. 

Between-group differences in summer zBMI gain were statistically significant between children from suburban and urban schools −0.035 (95CI = −0.062, −0.008), such that children from urban schools had significantly greater zBMI gain when compared to children from suburban schools. There were also significant differences in summer zBMI change between exurban and urban schools’ zBMI −0.037 (95CI = −0.060, −0.014), such that summer acceleration was significantly greater among urban children compared to children from exurban schools.

Between-group differences were not statistically significant for children from suburban schools during the summer months for %BMIp95 −0.002 (95CI = −0.005, 0.001), such that children from urban schools’ difference in summer accelerations were not significantly different when compared to children from suburban schools. Further, between-group differences in the slope of the linear equation were statistically significant for children from exurban schools during the summer months for %BMIp95 −0.002 (95CI = −0.005, 0.000), indicating that children from urban schools’ difference in summer accelerations more significant when compared to children from exurban schools. 

## 4. Discussion

This study sought to examine differences in accelerations of children’s BMI during the summer months by school locality. Overall, children’s BMI, zBMI, and %BMIp95 gain was statistically significantly greater during the summer months when compared to the school year. Across BMI and %BMIp95, children from urban schools experienced statistically significant accelerations during the summer months compared to the school year. Children from urban school’s summer accelerations in zBMI appear to be statistically significantly greater when compared to children from suburban and exurban designated schools. Like zBMI, children from urban schools experienced significantly greater accelerations in %BMIp95 during the summer months compared to their exurban counterparts. These findings contradict past work that has found children from urban areas are less likely to be overweight/obese compared to children from exurban regions [46,47]. 

Reasons for these contradictory findings could be the unbalanced nature of the sample. First, in this sample, a higher proportion of children who reside in exurban settings were from high-income households (>2.0 PIR) compared to urban settings (37% vs. 31%), respectively. These differences in PIR could be contributing to the differences in summer acceleration presented herein. The Health Gap Hypothesis states that children from low-income households experience a greater amount of weight gain during the summer when compared to their high-income peers because children from low-income households have less access to community programming during the summer [48]. High-income children in exurban areas may not be affected by the distance and transportation barriers that low-income children in exurban areas experience. Thus, this may explain why they did not experience accelerated summer BMI gain at the same rate as their urban peers in this sample. Second, in this sample, a significantly higher proportion of children who reside in urban settings are from minority households compared to exurban settings (76% vs. 40%), respectively. These differences in race/ethnicity could also explain the findings of this study, and could be considered confounding. Studies have shown that children from minority households are more susceptible to accelerated summer BMI gain [26]. This may be because children from minoritized racial groups are more likely to live in lower-resourced communities and experience unique stressors due to systemic racism [49]. This may explain the relatively higher rates of accelerated BMI gain in the urban children in the current sample. Finally, utilizing BMI as the primary outcome measure may be contributing to our findings. In this sample, White children make up the largest proportion of exurban children compared to Black (60% vs.10%). Studies have shown that BMI is a poor indicator of excess body fat in Black or African American children. Between 30–60% of Black children are misclassified as overweight or obese when they do not have excess body fat [50,51,52].

Future studies of summer changes in body composition should include complementary measures of body composition (e.g., bioelectrical impedance) beyond BMI. Adding these additional measures may help to understand better yearly trajectories of body composition given the limitations with BMI. BMI for youth is widely used as a surveillance and outcome measure of body composition in clinical trials because it is non-invasive, fast, and easy to employ for weight status screening (i.e., underweight, normal or healthy weight, overweight, and obesity), and can be readily deployed in field-based research settings where measures of hundreds, if not thousands, of youth, may occur (e.g., schools) [53]. While evidence has shown that BMI is moderately correlated with children’s body composition [54,55,56], it provides no information on an individual’s body composition. Rather, BMI is a simple ratio of an individual’s height to weight. This can be problematic because individuals with the same BMI may have dramatically different body compositions. For instance, recent studies have demonstrated BMI’s inability to distinguish fat from lean mass, which can lead to inappropriate diagnosis of excess adiposity in up to 25% of children [57,58]. Further, changes in children’s BMI as they mature may be primarily driven by gains in fat-free mass (FFM), not adiposity, especially for boys [59]. Because of these limitations, supplementing measures of BMI with non-invasive measures of body composition could better inform efforts to assess the prevalence of and target intervention efforts towards addressing childhood obesity, all while producing greater certainty in measures of body composition in research studies. 

The current study included several strengths. First, after restricting the sample, a large final sample is still included in analysis. Second, our study included the use of BMI, zBMI, and %BMIp95 to illustrate the differences and weaknesses provided by all three measures. Third, restricting the sample to those measured in August/September and April/May allowed us to explicitly examine school and summer changes without predicting the measures of children assessed outside these measurement windows. Finally, incorporating BH to account for Type-I error at a 10% false discovery rate allowed multiple comparisons. After correcting for multiple comparison testing, all findings that originally reached statistical significance (*p* < 0.05) remained in the post-hoc analysis. This study is also not without its limitations. ECLS-K:2011 collected baseline data in 2010–2011, thus utilizing data this old may be considered a weakness. Further, the effects and clinical significance may be questioned as some BMI findings did reach statistical significance, although the magnitude and effect size of change were relatively small. Past studies have concluded that reductions in mean zBMI of 0.15 and BMI standard deviations of 0.7–1.2 are associated with significant improvements in lipid, insulin, blood pressure, total cholesterol, and low-density lipoprotein [60,61]. The findings presented herein are presented as monthly changes. When examining yearly trajectories and monthly changes in measures of BMI, it should be noted that small differences on a monthly level may be important as they compile over time. To continue, multiple summers of slightly higher accelerations may be contributing to health outcomes over time.

## 5. Conclusions

BMI gain accelerated during the summer months when compared to the school year. Between-group differences indicated that BMI accelerates at a greater rate for children from urban schools during the summer months than suburban and exurban children. Interventions are needed that specifically address obesogenic behaviors during the summer months, especially for children in urban settings, who are more likely to be low-income and from racially minoritized groups.

## Figures and Tables

**Figure 1 ijerph-18-11610-f001:**
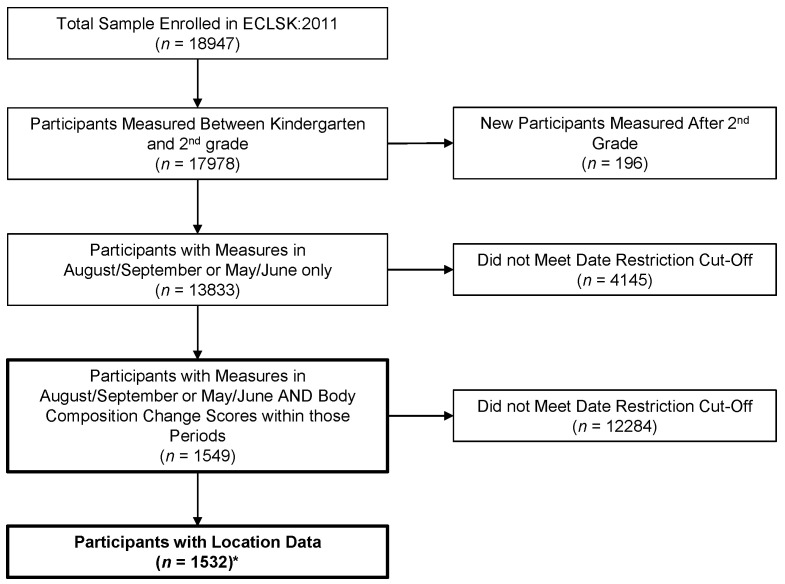
Flow Diagram of Participants Included. * indicates final sample included in analysis.

**Table 1 ijerph-18-11610-t001:** Demographics of Analytical Sample and Full Sample.

	Analytical Sample (N = 1532)						Full Sample (N = 18,947)					
	Urban (*n* = 550)		Suburban (*n* = 468)		Exurban (*n* = 514)		Urban (*n* = 6414)		Suburban (*n* = 6895)		Exurban (*n* = 5638)	
Income, n (%)												
* Low (0.0–0.9 PIR)	132	24%	97	21%	101	20%	1642	26%	1113	16%	917	16%
Med (1.0–1.9 PIR)	83	15%	66	14%	97	19%	1098	17%	1061	15%	1007	18%
* High (>2.0 PIR)	173	31%	183	39%	192	37%	1786	28%	3179	46%	2371	42%
Missing	162	29%	126	26%	124	24%	1888	29%	1542	22%	1343	24%
Mean Age (±SD)	7.13	0.76	6.93	0.88	6.71	0.69	6.95	0.98	6.98	1.00	7.01	1.01
Mean BMI (±SD)	17.22	3.20	17.15	3.11	17.00	3.22	17.22	3.16	17.03	2.96	17.14	3.04
zBMI (±SD)	0.5055	1.10	0.5381	1.06	0.5107	1.08	0.5387	1.12	0.4740	1.09	0.5194	1.07
Sex, *n* (%)												
Boys	284	52%	241	51%	273	53%	3209	50%	3530	51%	2952	52%
Girls	266	48%	227	49%	241	47%	3192	50%	3350	49%	2674	47%
Race, *n* (%)												
* White	134	24%	163	35%	308	60%	1681	26%	3197	46%	3830	68%
* Black	79	14%	36	8%	49	10%	1256	20%	784	11%	475	8%
* Hispanic	236	43%	204	44%	101	20%	2219	35%	1895	27%	811	14%
* Asian	59	11%	43	9%	13	3%	839	13%	573	8%	187	3%
* Other	38	7%	20	4%	43	8%	403	6%	420	6%	329	6%
Missing	4	1%	2	0%	0	0%	16	0%	26	0%	6	0%
Parent Employed, *n* (%)												
Yes	278	51%	282	60%	333	65%	2492	39%	3099	45%	2791	50%
No	171	31%	209	45%	167	32%	1789	28%	1954	28%	1639	29%
Missing	1	0%	0	0%	14	3%	2133	33%	1842	27%	1208	21%
Parents Highest Education, *n* (%)												
High School or Less	147	27%	133	28%	164	32%	1761	27%	1394	20%	1466	26%
Some College	52	9%	58	12%	72	14%	742	12%	856	12%	882	16%
Associates	18	3%	23	5%	46	9%	320	5%	468	7%	451	8%
Bachelors	77	14%	90	19%	76	15%	793	12%	1378	20%	915	16%
Greater than Bachelors	41	7%	48	10%	41	8%	415	6%	723	10%	397	7%
Other	10	2%	18	4%	20	4%	242	4%	236	3%	314	6%
Missing	205	37%	98	21%	95	18%	2141	33%	1840	27%	1213	22%

BMI = Body Mass Index, SD = standard deviation, PIR = Poverty-to-Income Ratio; Chi-Square was implemented to determine if relationship exist between income and location and race/ethnicity and location. ***** indicates *p* < 0.05.

**Table 2 ijerph-18-11610-t002:** Mixed-Effect Linear Regression Models Estimating Difference in Monthly Change during the School Year and Summer.

	Raw School Change			Raw Summer Change			Model Based within Group Difference			Model Based between Group Diff in Change		
BMI												
	*n*	Mean	SD	*n*	Mean	SD	Coef.	95CI		Coef.	95CI	
Full Sample	1214	0.028	0.193	710	0.071	0.230	* 0.095	0.074	0.117			
Urban	415	0.030	0.157	277	0.078	0.247	* 0.083	0.049	0.117	ref		
Suburb	369	0.027	0.158	213	0.088	0.202	* 0.047	0.004	0.090	−0.036	−0.090	0.018
Exurban	398	0.028	0.256	196	0.053	0.240	* 0.039	0.009	0.070	−0.044	−0.090	0.002
zBMI												
Full Sample	1209	0.002	0.074	710	0.007	0.126	* 0.016	0.004	0.027			
Urban	414	−0.001	0.074	277	0.014	0.140	0.043	0.026	0.060	ref		
Suburb	367	0.006	0.076	213	0.012	0.098	0.008	−0.013	0.029	* −0.035	−0.062	−0.008
Exurban	396	0.002	0.075	196	−0.004	0.139	0.006	−0.009	0.021	* −0.037	−0.060	−0.014
Percent of 95 Percentile Change												
Full Sample	1214	−0.001	0.010	710	0.000	0.012	* 0.001	0.000	0.002			
Urban	415	−0.002	0.008	277	0.001	0.013	* 0.004	0.002	0.006	ref		
Suburb	369	−0.001	0.008	213	0.001	0.010	0.002	−0.000	0.004	−0.002	−0.005	0.001
Exurban	398	−0.001	0.013	196	−0.001	0.012	0.002	0.000	0.003	−0.002	−0.005	0.000

School and summer change estimates are presented as raw means and standard deviations. All models included sex, age, parent’s employment status, parent’s education, race, and poverty-to-income ratio as covariates. Within and between-group difference estimates are model-based. Benjamini–Hochberg post-hoc analysis accounted for type-I error set at 10% false discovery rate. Values * represent *p* < 0.05.

## Data Availability

All publicly available data used in this study can be found here: https://nces.ed.gov/ecls/datainformation2011.asp (accessed on 6 November 2019).

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
