# Peer review of "Differences by School Location in Summer and School Monthly Weight Change: Findings from a Nationally Representative Sample"

_ijerph, 2021, doi:10.3390/ijerph182111610_

Round 1

Reviewer 1 Report

This research paper compares regional differences (urban vs. rural)  and examines whether there is a difference in the degree of obesity progression in the school season and the summer period.

 The statistical methods are sound, and the study is original, especially in its use of %BMIp95. In reviewing the paper, I think the following two points need to be corrected to make the paper more complete, so I propose a minor revise.

1) Background is listed in Table 1, but it needs to be clearly stated whether there is a statistically significant difference between groups in the region. The discussion states that there are large differences in income, minority, and race between urban and rural areas (L317-L340), but if there is a significant difference, it could be a confounding factor with regional differences, so the results should be listed in table 1. If there is a significant difference, the discussion should include "These factors in Table 1 may be confounders. If there is a significant difference, the discussion should include "These factors in Table 1 may be confounding factors."

2) Are the urban results for %BMIp95 listed in Table 2 and the results reproducible in other studies? For example, when the best linear equation is calculated for each region with the rate of increase in BMI during the school year and the summer vacation period, is the slope of the linear equation steeper only in the cities during the summer? If you have considered this, it would be more convincing if you add a line to the results, even just one line.

Author Response

We thank the reviewer for their thoughtful comments and positive view of our work. We believe that addressing the reviewer comments has strengthened this manuscript considerably.  Our responses to reviewer comments are in underlined bolded text below. Changes to the original manuscript are indicated by highlighted yellow text.

Reviewer 1

This research paper compares regional differences (urban vs. rural)  and examines whether there is a difference in the degree of obesity progression in the school season and the summer period.

The statistical methods are sound, and the study is original, especially in its use of %BMIp95. In reviewing the paper, I think the following two points need to be corrected to make the paper more complete, so I propose a minor revise.

1) Background is listed in Table 1, but it needs to be clearly stated whether there is a statistically significant difference between groups in the region. The discussion states that there are large differences in income, minority, and race between urban and rural areas (L317-L340), but if there is a significant difference, it could be a confounding factor with regional differences, so the results should be listed in table 1. If there is a significant difference, the discussion should include "These factors in Table 1 may be confounders. If there is a significant difference, the discussion should include "These factors in Table 1 may be confounding factors."

This comment is appreciated by the authors and has been addressed. Significant differences did exist between income and location, as well as race/ethnicity and location. The test of significance has been added to table 1 as recommended. In addition, a section highlighting this as a potential confounder has been added to the discussion section.

Page 8, line 330. Second, in this sample, a significantly higher proportion of children who reside in urban settings are from minority households compared to exurban settings (76% vs. 40%), respectively. These differences in race/ethnicity could also explain the findings of this study and could be considered confounding.

2) Are the urban results for %BMIp95 listed in Table 2 and the results reproducible in other studies? For example, when the best linear equation is calculated for each region with the rate of increase in BMI during the school year and the summer vacation period, is the slope of the linear equation steeper only in the cities during the summer? If you have considered this, it would be more convincing if you add a line to the results, even just one line.

A line addressing the slope of the linear equation for %BMIp95 (percent of the 95th percentile change) has been added to the results.

Page 7, line 295. Further, between-group differences in the slope of the linear equation were statistically significant for children from exurban schools during the summer months for %BMIp95 -0.002 (95CI= -0.005, 0.000), indicating that children from urban schools' difference in summer accelerations more significant when compared to children from exurban schools.

Reviewer 2 Report

General comments: Thank you for the opportunity to review this manuscript title “Differences by School Location in Summer and School Monthly Weight Change: Findings from a Nationally Representative Sample”. First of all, I congratulate you for your work. I find this study very interesting. 
However, please see my comments below:

Abstract: 
- page 1 line 14: There is a mistake in "Kindergarten Class years 2010-201". Add the last number (number 1) in the year 2011.

Introduction:
- page 2 lines 78-80: “Previous studies have shown that Black children and children from low-income House...”. Terms such as “previous studies” need to be supported with more than just one reference. Please provide the references or change this term. 

Results:
- page 7 line 281 and table 2: In the following results “0.002 (95CI= -0.000, 0.004)”, you added the negative sign before 0.000. However, this sign is not in table 2. Could you add it in table 2?

Author Response

We thank the reviewer for their thoughtful comments and positive view of our work. We believe that addressing the reviewer comments has strengthened this manuscript considerably.  Our responses to reviewer comments are in underlined bolded text below. Changes to the original manuscript are indicated by highlighted yellow text.

General comments: Thank you for the opportunity to review this manuscript title “Differences by School Location in Summer and School Monthly Weight Change: Findings from a Nationally Representative Sample”. First of all, I congratulate you for your work. I find this study very interesting. 
However, please see my comments below:

Abstract: 
- page 1 line 14: There is a mistake in "Kindergarten Class years 2010-201". Add the last number (number 1) in the year 2011.

This comment has been addressed in the revised version on page 1 line 14 and is highlighted in yellow.

Introduction:
- page 2 lines 78-80: “Previous studies have shown that Black children and children from low-income House...”. Terms such as “previous studies” need to be supported with more than just one reference. Please provide the references or change this term. 

The below additional reference has been added and is highlighted in yellow.

von Hippel, P. T., & Workman, J. (2016). From kindergarten through second grade, US children's obesity prevalence grows only during summer vacations. Obesity, 24(11), 2296-2300.

Results:
- page 7 line 281 and table 2: In the following results “0.002 (95CI= -0.000, 0.004)”, you added the negative sign before 0.000. However, this sign is not in table 2. Could you add it in table 2?

This comment has been addressed in the revised version page 7 line 281 and table 2 and is highlighted in yellow.

Reviewer 3 Report

Regarding the originality of the study, it does not seem original to me, although it is interesting. I fully agree that where children live and attend school can contribute significantly to overweight and obesity.

That the body mass index (BMI) of children accelerates and the prevalence of children with obesity increases during the summer months is well known.

The keywords school location and summer are already in the title. These must be replaced.

This review has carried out an analytical sample of 1549 subjects, which I consider appropriate. The problem is that the sample of the study on which it is based is from 2010-11

In the measurements studied (weight and height) the authors specify that they were performed twice for each occasion. Nothing specify the authors that value was taken as final. In anthropometry the values ​​are taken three times, the final value being the median or the mode.

I agree that future studies of summer changes in body composition should include complementary measures of body composition (Ej., anthropometry or Bioelectrical impedance) beyond BMI.

The main problem with this study, as stated by the authors, is that BMI is not the appropriate indicator to determine body composition. The same BMI can have very different body compositions.

The data analysis is adequate.

I do not consider that the results provide an advance in the current knowledge of the subject.

The conclusions are supported by the results although they are contradictory to the scientific literature. “Children living in exurban settings are significantly more likely to be overweight / obese compared to their urban counterparts [5,9]. A recent meta-analysis concluded that children (2 to 19 years) in exurban settings are more than 25% more likely to be obese compared to urban children [6] ”.

The bibliography seems adequate to me

Author Response

The authors thank the reviewers for their comments regarding this manuscript. All comments have been addressed and have been added to a revised version of the manuscript and highlighted in yellow.

Reviewer 3

Regarding the originality of the study, it does not seem original to me, although it is interesting. I fully agree that where children live and attend school can contribute significantly to overweight and obesity.

We have highlighted the unique contributions of this study in our strengths section (Page 9 line 363). While other studies have established that place can influence health, no other study up to this point has examined summer change by locality. Johnson et al. concluded that exurban children have more than 25% greater odds of obesity compared to urban children.1 Because of this, we believe this study has its merit as it begins to examine when changes in body composition are occurring by locality. 

That the body mass index (BMI) of children accelerates and the prevalence of children with obesity increases during the summer months is well known.

This is correct. However, to our knowledge, this is the first to examine the summer weight gain phenomenon by locality.

The keywords school location and summer are already in the title. These must be replaced.

This comment has been addressed in the revised version, and we have removed the keywords school location and summer.

This review has carried out an analytical sample of 1549 subjects, which I consider appropriate. The problem is that the sample of the study on which it is based is from 2010-11.

The authors understand that a study utilizing baseline data from 2010-2011 may be a weakness. We have added this as a weakness in the discussion.

Page 9, line 371. ECLS-K:2011 collected baseline data in 2010-2011, thus utilizing data this old may be considered a weakness.  

In the measurements studied (weight and height) the authors specify that they were performed twice for each occasion. Nothing specify the authors that value was taken as final. In anthropometry the values ​​are taken three times, the final value being the median or the mode.

ECLK-K:2011 collected measures twice on each occasion. Because of this, the average of the two values was used. To clarify, we added this to the methods section.

Page 3, line 122. BMI was calculated by first taking the average of the measures, then using the standard formula BMI=kilograms/meter.

I agree that future studies of summer changes in body composition should include complementary measures of body composition (Ej., anthropometry or Bioelectrical impedance) beyond BMI.The main problem with this study, as stated by the authors, is that BMI is not the appropriate indicator to determine body composition. The same BMI can have very different body compositions.

We mention in the discussion section (Page 9, Lines 337-341) that studies have shown that BMI is a poor indicator of excess body fat in Black or African American children. Because of this, we agree that complementary measures of body composition should be utilized.

The data analysis is adequate.

Analysis was performed using STATA v16.

I do not consider that the results provide an advance in the current knowledge of the subject.

The authors believe that findings from the current futher illucidate reasons for the summer weight gain phenomenon. Future projects should be designed to explore the summer weight phenomenon by location specifically.

The conclusions are supported by the results although they are contradictory to the scientific literature. “Children living in exurban settings are significantly more likely to be overweight / obese compared to their urban counterparts [5,9]. A recent meta-analysis concluded that children (2 to 19 years) in exurban settings are more than 25% more likely to be obese compared to urban children [6] ”.

The authors believe that because our findings contradict the scientific literature, this research question needs to be replicated in a study that explicitly examines the summer weight gain phenomenon by location.

The bibliography seems adequate to me

References were cited and organized using Endnote referencing software

Reviewer 4 Report

Excellent paper but I have two suggestions for you to consider:

(1) the abstract is really dense with data and hard to read so you might consider shortening the abstract and making it more readable and

(2) Table 2 in the first section appears to be missing a bolded heading and there seems to be an extra space between Full Sample and Urban data sections. 

Author Response

We thank the reviewer for their thoughtful comments and positive view of our work. We believe that addressing the reviewer comments has strengthened this manuscript considerably.  Our responses to reviewer comments are in underlined bolded text below. Changes to the original manuscript are indicated by highlighted yellow text.

Reveiwer 4

Excellent paper but I have two suggestions for you to consider:

  • the abstract is really dense with data and hard to read so you might consider shortening the abstract and making it more readable.

This comment is appreciated as there is more data in the abstract than we would usually like to see. However, we respectfully believe that the data presented is warranted in the abstract and is within the journal’s word limits.

(2) Table 2 in the first section appears to be missing a bolded heading and there seems to be an extra space between Full Sample and Urban data sections. 

This comment has been addressed in the revised version and is highlighted in yellow.

References cited in response.

  1. Johnson III JA, Johnson AM. Urban-rural differences in childhood and adolescent obesity in the United States: a systematic review and meta-analysis. Childhood obesity. 2015;11(3):233-241.

This manuscript is a resubmission of an earlier submission. The following is a list of the peer review reports and author responses from that submission.